# The Beauty of Simplicity: Delayed-Type Hypersensitivity Reaction to Measure Cellular Immune Responses in RNA-SARS-Cov-2 Vaccinated Individuals

**DOI:** 10.3390/vaccines9060575

**Published:** 2021-06-01

**Authors:** Yvelise Barrios, Andres Franco, Inmaculada Sánchez-Machín, Paloma Poza-Guedes, Ruperto González-Pérez, Victor Matheu

**Affiliations:** 1Department of Immunology, Hospital Universitario de Canarias, 38320 San Cristóbal de La Laguna, Spain; ybarpin@gobiernodecanarias.org (Y.B.); aframas@gobiernodecanarias.org (A.F.); 2Department of Allergy, Hospital Universitario de Canarias, 38320 San Cristóbal de La Laguna, Spain; msanmack@gobiernodecanarias.org (I.S.-M.); ppozgue@gobiernodecanarias.org (P.P.-G.); rgonperl@gobiernodecanarias.org (R.G.-P.)

**Keywords:** SARS-CoV-2, DTH, skin test, delayed-type hypersensitivity, T-cell response, COVID-19, cell response, humoral response

## Abstract

Background: Monitoring cellular immune responses elicited in vaccinated individuals is highly complicated. Methods: 28 individuals participated during the vaccination process with 12 BNT162b2 mRNA (Pfizer) vaccine. Specific anti-RBD IgG using a classic ELISA was performed in days 10 and 20 (after one dose of the vaccine) and on day 35 (after two vaccine doses) in serum samples of all participants. In parallel, DTH (delayed-type hypersensitivity) Skin Test using S protein was performed before (11/28) and after two doses (28/28) of the vaccine. Results: 6/28 individuals were considered positive for the specific anti-RBD IgG positive at day 10, whereas all 28 individuals were positive at day 20. Moreover, 28/28 individuals increased the OD ratios at day 36 (2 doses). DTH cutaneous test was performed on 11/28 participants at day 20 (1 dose) showing 8/11 a positive reaction at 12 h. DTH of all participants was performed on day 36 (2 doses), showing 28/28 positive reactions at 12 h. Conclusion: This report describes the first publication of the results obtained using an in vivo method, the classical DTH response to the Spike protein to assess T-cell immune responses in vaccinated individuals. This affordable and simple test would help to answer basic immunogenicity questions on large-scale population vaccine studies.

## 1. Introduction

Ongoing studies are monitoring immune responses elicited by mRNA vaccines. Few studies have been focused on the role that the immune cellular responses might play in the development of immunity to SARS-CoV-2 and what are the implications for vaccines. Usually, most studies have systematically explored only humoral responses assuming that individuals with high antibody levels after infection should also have a high number of SARS-CoV-2 specific T cells. However, there is evidence that T cells producing interferon γ have also been detected a median of 75 days after PCR confirmed COVID-19 in people with undetectable SARS-CoV-2 antibodies [1], suggesting immunity is partly mediated and maintained by memory T cells. Finally, a preprint of a recent study of 100 people with a history of asymptomatic or mild COVID-19 reports T cell mediated immune responses lasting for at least six months in all participants [2].

Most of these studies have been mainly focused on the humoral side with few studies addressing cellular immune responses [3,4,5]. One of the reasons behind this lack of studies is that both ELISAs antibody methods and in vitro cellular assays require the extraction of a blood sample from the patient that complicate massive analysis in large populations. Here, we present the results of a novel application of an in vivo method to monitor cellular immune responses elicited in vaccinated individuals. This method has been previously validated in SARS-CoV-2 exposed individuals [6] showing a strong concordance (84.3%) with anti-RBD IgG.

## 2. Materials and Methods

### 2.1. Participants

Twenty-eight individuals were enrolled from the health care workers priority group vaccinated at the Hospital Universitario de Canarias during February–March 2021. These patients were to be vaccinated with the BNT162b2 mRNA Pfizer vaccine. All of them were given a written document with the relevant and necessary information to decide on their participation in the study. The study is conducted in accordance with the requirements expressed in Law and the protocol was approved by the ethical committee of the Hospital (CHUC_2021_04).

### 2.2. Serology

The vaccine administration schedule was carried out on days 0 and 21. Serum samples were collected from the participants on days 0, 10, 20 (just before the second dose) and on day 36 (15 days after the second dose). Every single sample was sent to the Immunology laboratory and frozen. and sent to the Immunology laboratory. A commercial ELISA IgG specific for the S1 protein of SARS-CoV-2 was used according to manufacturer’s instructions (*Euroimmun*, Lübeck, Germany). Results were expressed as Optical Density (O.D.) ratios as it has been recently described and validated for these determinations [7]. OD ratios under 0.8 were considered negative.

### 2.3. Delayed-Type Hypersensitivity (DTH) Skin Test

The protocol was performed according to usual clinical practice [8] and as previously described [4]. Briefly, after signing the informed consent, 25 microL (0.1 mg/mL final concentration) for intradermal puncture (IDT) of the spike protein was administered in the volar part of arm. A lyophilized recombinant SARS-CoV-2 protein of the receptor binding domain (RBD) was re-suspended in sterile water and sterile filtered 0.22 µm with a final concentration of 0.1 mg/mL following the manufacturer’s instructions and under controlled sterilization conditions. The final concentration was obtained from that normally used in the tuberculin test [9]. The participants had been instructed to avoid antihistamines and corticosteroids at least 5 days before the DTH skin test and to take a well-focused photograph with a ruler to assess the size of the reaction at 6 h, 12 h, 24 h, and 48 h after injection. DTH was performed in all patients on day 36, 14 days after the second dose. A 24-h helpline was provided for their consultation and evaluation if necessary. A positive response was considered in the case of a positive cellular response function. Additionally, DTH was also performed in 11 patients on day 20. Ten negative controls without previous infection and without vaccination were used to see the null responsiveness of the protein.

## 3. Results

### 3.1. Specific IgG anti Spike

We evaluated immune responses in 28 immunocompetent subjects who received two doses of BNT162b2 mRNA Pfizer vaccination. Six out of 28 sera collected at day 10 showed specific anti-RBD IgG OD ratio values considered positive (>0.8), whereas serum from each of 28 individuals were positive at day 20 (1 dose). Moreover, all 28 individuals increased the OD ratios at day 36 (2 doses) (Figure 1).

### 3.2. Delayed-Type Hypersensitivity Test

Skin DTH test was performed on day 36, 14 days after second dose, in all 28 individuals and showing 28/28 positive reactions. Mean diameter was 8.12 mm (STD 3.54) after 6 h of puncture, 14.43 mm (6.00) after 12 h, 18.29 mm (6.51) after 24 h and 14.18 mm (6.06) after 48 h of puncture (Table 1) (Figure 2a). Additionally, DTH cutaneous test had been performed on 11 participants at day 20, prior to the second dose. Results showed that 8 out of 11 individuals had a positive reaction (Figure 2b). Mean diameter of DTH skin test in the eight positive individuals was 3.2 mm after 6 h of puncture, 6.63 mm (5.26) after 12 h, 13.25 mm (4.71) after 24 h and 13 mm (8.39) after 48 h of puncture. No individual exhibits an immediate reaction after 15 min and up to 45 min after the IDT administration.

## 4. Discussion

Nucleic acid (DNA or RNA) vaccines are considered the next generation vaccines as they can be rapidly designed to encode any desirable viral sequence including the highly conserved antigen sequences. RNA vaccines being less prone to host genome integration and anti-vector immunity (a compromising factor of viral vectors) offer great potential as front-runners for universal COVID-19 vaccine. The proof of concept for RNA-based vaccines has already been proven in humans with safety and immunogenicity outcomes that confirms these high expectations [10].

As it has been previously reported, we found that although all participants were anti-RBD IgG positive before the second dose, there is a remarkable increase in the OD ratio on day 36 (after 2 doses). However, the skin DTH test was positive in 8/11 individuals after a single dose of vaccine. All participants (28/28) developed a positive DTH test after two doses of the vaccine, showing that It seems that both vaccine doses are needed for the detection of in vivo T cell immune reaction. This finding would be in line with the in vitro data previously reported about the presence of memory B cells specific for the spike receptor binding domain (RBD) efficiently primed by mRNA vaccination and detectable after the second vaccine dose [3]. The kinetics of DTH skin reaction was more rapid (12 h) and wanes faster (48 h) in vaccinated individuals compared with natural immunized patients [4].

DTH responses are a component of the type IV hypersensitivity reaction category of cell-mediated immunity. Unlike types I–III, which involve various forms of antibody-mediated activities, only effector T cells and activated macrophages participate in DTH responses [11]. These responses are often associated with the host response to intracellular pathogens [12]. Their detection is best exemplified by the well-known tuberculin test (also known as the Mantoux or purified protein derivative [PPD] skin test) [9]. When a patient previously exposed to Mycobacterium tuberculosis is injected with a small amount of PPD (tuberculin) intradermally, there is little reaction in the first few hours. Gradually, however, induration and erythema develop which reaches a peak at 48–72 h. A positive skin test indicates that the person has been infected with the agent, but it does not necessarily confirm the presence of current disease. Cell-mediated immunity/hypersensitivity also develops in many viral infections including mumps, herpes and, to some extent, measles. We have use in this report the DTH test to assess the cellular immune responses elicited by mRNA vaccines to COVID-19. [13]

Whereas the current successful human antiviral vaccines, such as influenza and measles vaccines, depend largely on the induction of antibody responses, emerging evidence suggests the requirement of both antibody-mediated and T cell-mediated immunity for effective protection against SARS-CoV-2 [14,15]. These lines of evidence, together with data suggesting that T cell-mediated immunity generally is a more reliable correlate of vaccine protection than antibody titers in seniors [16], strongly support the inclusion of T cell responses analysis in COVID-19 vaccine design. However, there are several technical and logistical inconvenient to perform these sophisticated cellular immune studies in large populations [17].

To the best of our knowledge, this report describes the first publication of the results obtained using an in vivo method, the classical delayed-type hypersensitivity (DTH) response to the intradermal injection of the Spike protein to assess T-cell immune responses in vaccinated individuals. This affordable and simple test would help to answer basic immunogenicity questions on large-scale population vaccine studies.

## 5. Conclusions

This report describes the first publication of the results obtained using an in vivo method, the classical delayed-type hypersensitivity (DTH) response to the intradermal injection of the Spike protein to assess T-cell immune responses in vaccinated individuals. This affordable and simple test would help to answer basic immunogenicity questions on large-scale population vaccine studies.

## 6. Patents

Y.B and V.M. have filed (79241/P8547) provisional Utility Model applications related to DTH tests for cellular immunity against SARS-CoV-2.

## Figures and Tables

**Figure 1 vaccines-09-00575-f001:**
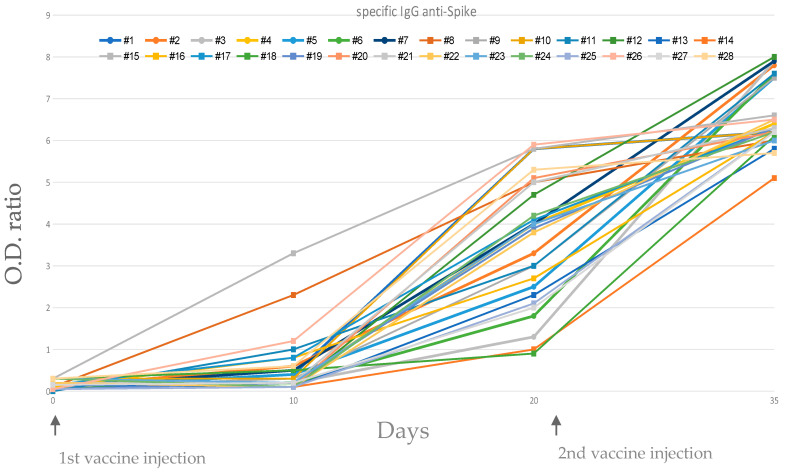
Evolution of IgG antibody titers against SARS-CoV-2 spike protein in individuals. Sera were obtained on days 0, 10, 20 (after first dose of vaccine) and 35 (2 weeks after second dose of vaccine).

**Figure 2 vaccines-09-00575-f002:**
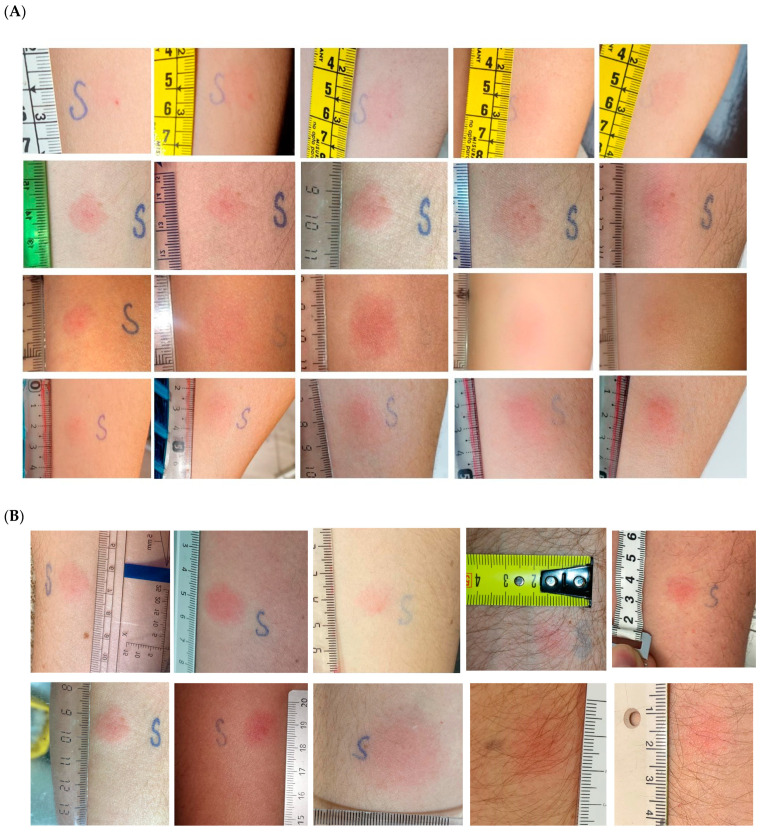
**Delayed-type hypersensitivity cutaneous test.** (**A**). Representative image of DTH skin test in four immunocompetent individuals showing reaction (First line: 35y-o woman; Second line: 51y-o woman; Third line: 48 y-o woman; Fourth line 49y-o. woman) at 6 h (first column), 12 h (second column), 24 h (third column), 36 h (fourth column), and 48 h (fifth column). (**B**). Representative image of DTH skin test in five immunocompetent individuals. Images show in the first line the skin responses 12 h after DTH performed twenty days after first dose. In second line images show the skin reaction 12 h after DTH performed fifteen days after second dose.

**Table 1 vaccines-09-00575-t001:** Diameter of DTH skin reaction in all 28 individuals at 6 h, 12 h, 24 h, and 48 h after intradermal test puncture according to the reports of the photographic impressions sent by the patients.

Hour	1	2	3	4	5	6	7	8	9	10	11	12	13	14	15	16	17	18	19	20	21	22	23	24	25	26	27	28
**6**	6	14	14	8	10	6	6	8	14	10	8	10	0	8	8	6	9	11	7	12	14	6	6	7	8	0	6	8
**12**	12	12	30	6	22	8	9	10	20	22	16	24	4	16	12	15	15	20	16	14	22	12	12	9	12	9	9	16
**24**	20	13	15	7	30	6	12	13	22	30	17	29	10	22	26	22	19	16	20	19	27	14	14	22	22	15	12	18
**48**	8	14	15	14	8	2	18	9	22	20	20	11	12	20	10	16	34	16	14	14	16	12	12	6	14	14	18	8

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
