# Peer review of "The Beauty of Simplicity: Delayed-Type Hypersensitivity Reaction to Measure Cellular Immune Responses in RNA-SARS-Cov-2 Vaccinated Individuals"

_vaccines, 2021, doi:10.3390/vaccines9060575_

Round 1
Reviewer 1 Report
Title:
It’s hypersensitivity, not “hipersensitivity” – misspelling in the title!
And what is “mRNA-SARS-CoV-2”? Do the authors mean “SARS-CoV-2 mRNA”? But viruses don’t have mRNA… Never seen this “mRNA-SARS-CoV-2” nomenclature before.
Abstract:
Please write one! Right now it’s just “A single paragraph of about 200 words maximum … should not exaggerate the main conclusions. “ placeholder!!!!!
Introduction:
“One of the reasons behind this lack of studies is that both ELISAs antibody methods“ – isn’t that looking at humoral response, antibodies??
Results:
“BNT162b2 vaccine” – please state which brand vaccine this is – Pfizer or Moderna?
Figure 1: The graph shows OD ratio, clearly positive since the threshold but what does it mean in terms of amount of anti-RBD? Usually one measures OD but then has a calibration curve so in the final figure what should up is the amount of anti-RBD IgG, not the raw OD values or ratios…
Figure 2.
Please delete “This is a figure.”
Discussion
“All participants (28/28) developed a positive DTH test after two doses of the vaccine, showing that It seems that both vaccine doses are needed for the detection of in vivo T cell immune reaction.”
- The authors have to explain this better. It’s very obvious that they collect serum and do some colorimetric assay to measure amount of anti_RBD IgG in the samples (which as mentioned above is not specified how much, it jut shows that the OD goes up), but an in vivo assay is more complex than that, so please explain briefly the immunology behind injecting antigen intradermally and the rash appearance and how that’s indicative of a T cell response.
Patents:
“All authors declare that they have no competing interests.” – this can’t be true if in the previous sentence before the authors stated that some of them filed a patent on this. Maybe they mean “All OTHER authors…”?
Author Response
Reviewer #1
Title: It’s hypersensitivity, not “hipersensitivity” – misspelling in the title!
We are sorry that we made that mistake. We have corrected it.
And what is “mRNA-SARS-CoV-2”? Do the authors mean “SARS-CoV-2 mRNA”? But viruses don’t have mRNA… Never seen this “mRNA-SARS-CoV-2” nomenclature before.
We regret again that we made that mistake. We have corrected it.
Abstract:
Please write one! Right now it’s just “A single paragraph of about 200 words maximum … should not exaggerate the main conclusions. “ placeholder!!!!!
We have added the abstract that we do not know why it was not previously entered by us. Thank you for your interest in reviewing the manuscript without an abstract available as we know of this added difficulty.
Introduction:
“One of the reasons behind this lack of studies is that both ELISAs antibody methods“ – isn’t that looking at humoral response, antibodies??
We have reconstructed the introduction so that it is much more clarifying of the objectives of the study.
Results:
“BNT162b2 vaccine” – please state which brand vaccine this is – Pfizer or Moderna?
It has been precisely added that Pfizer was the vaccine used.
Figure 1: The graph shows OD ratio, clearly positive since the threshold but what does it mean in terms of amount of anti-RBD? Usually one measures OD but then has a calibration curve so in the final figure what should up is the amount of anti-RBD IgG, not the raw OD values or ratios…
Figure 2.
Please delete “This is a figure.”
It has been accurately corrected
Discussion
“All participants (28/28) developed a positive DTH test after two doses of the vaccine, showing that It seems that both vaccine doses are needed for the detection of in vivo T cell immune reaction.”
- The authors have to explain this better. It’s very obvious that they collect serum and do some colorimetric assay to measure amount of anti_RBD IgG in the samples (which as mentioned above is not specified how much, it jut shows that the OD goes up), but an in vivo assay is more complex than that, so please explain briefly the immunology behind injecting antigen intradermally and the rash appearance and how that’s indicative of a T cell response.
We have reconstructed the discussion to try to argue the validity of the DTH response and briefly explain its usefulness.
Nucleic acid (DNA or RNA) vaccines are considered the next generation vaccines as they can be rapidly designed to encode any desirable viral sequence including the highly conserved antigen sequences. RNA vaccines being less prone to host genome integration (cons of DNA vaccines) and anti-vector immunity (a compromising factor of viral vectors) offer great potential as front-runners for universal COVID-19 vaccine. The proof of concept for RNA-based vaccines has already been proven in humans with safety and immunogenicity outcomes that confirms these high expectations. [10]
As it has been previously reported, we found that although all participants were anti-RBD IgG positive before the second dose, there is a remarkable increase in the OD ratio on day 36 (after 2 doses). However, the skin DTH test was positive in 8/11 individuals after a single dose of vaccine. All participants (28/28) developed a positive DTH test after two doses of the vaccine, showing that It seems that both vaccine doses are needed for the detection of in vivo T cell immune reaction. This finding would be in line with the in vitro data previously reported about the presence of memory B cells specific for the spike receptor binding domain (RBD) efficiently primed by mRNA vaccination and detectable after the second vaccine dose [3]. The kinetics of DTH skin reaction was more rapid (12h) and wanes faster (48h) in vaccinated individuals compared with natural immunization patients [4]
DTH responses are a component of the type IV hypersensitivity reaction category of cell-mediated immunity. Unlike types I–III, which involve various forms of antibody-mediated activities, only effector T cells and activated macrophages participate in DTH responses [11]. These responses are often associated with the host response to intracellular pathogens [12]. Their detection is best exemplified by the well-known tuberculin test (also known as the Mantoux or purified protein derivative [PPD] skin test) [9]. When a patient previously exposed to Mycobacterium tuberculosis is injected with a small amount of PPD (tuberculin) intradermally, there is little reaction in the first few hours. Gradually, however, induration and erythema develop which reaches a peak at 48–72 hours. A positive skin test indicates that the person has been infected with the agent, but it does not necessarily confirm the presence of current disease. Cell-mediated immunity/hypersensitivity also develops in many viral infections including mumps, herpes and, to some extent, measles. We have use in this report the DTH test to assess the cellular immune responses elicited by mRNA vaccines to COVID-19. [13]
Whereas the current successful human antiviral vaccines, such as influenza and measles vaccines, depend largely on the induction of antibody responses, emerging evidence suggests the requirement of both antibody-mediated and T cell-mediated immunity for effective protection against SARS-CoV-2 [14,15]. These lines of evidence, together with data suggesting that T cell-mediated immunity generally is a more reliable correlate of vaccine protection than antibody titres in seniors [16], strongly support the inclusion of T cell responses analysis in COVID-19 vaccine design. However, there are several technical and logistical inconvenients to perform these sofisticated cellular immune studies in large populations [17].
To the best of our knowledge, this report describes the first publication of the results obtained using an in vivo method, the classical delayed-type hypersensitivity (DTH) response to the intradermal injection of the Spike protein to assess T-cell immune responses in vaccinated individuals. This affordable and simple test would help to answer basic immunogenicity questions on large-scale population vaccine studies.
Patents:
“All authors declare that they have no competing interests.” – this can’t be true if in the previous sentence before the authors stated that some of them filed a patent on this. Maybe they mean “All OTHER authors…”?
It has been precisely corrected
Reviewer 2 Report
The present short report is a clear, concise, and well-written manuscript. The introduction is relevant. The results are compelling. Authors should fix minor punctuation errors. Overall, the authors report the use of a simple in vivo method to monitor T-cell immune responses in mRNA-SARS-Cov-2 vaccinated individuals. This simple serum test reports a remarkable increase in antibody titer post 35 days of 2 doses of vaccination and all 28 subjects developed positive DTH reaction after 2 doses. This testing method could be easily used for large-scale population vaccine studies.
Authors have already validated the DTH skin test to study the cellular immune response in COVID19 patients in their recently published paper (Barrios et al. 2021). Here, in this short communication, authors have used the same DTH skin reaction method to 2 doses of BNT162b2 mRNA vaccine in the participants to assess cellular immune response. An understanding of cellular immune response is critical in controlling infection, particularly in the case of COVID19 pandemic. Because of its affordability and simplicity, DTH skin tests could be easily used to determine the immunogenicity of newly developed vaccines in large population studies.
Authors can improve their paper by checking on the following:
- Introduction can be improvised. In the beginning, authors can give an overview of mRNA-based vaccines, why mRNA vaccine platform is urgently needed and could be a promising alternative to conventional ones, further documenting its safety and efficacy.
- Materials and Methods, Skin Test- Here, authors should mention the injection site. In their cited reference, it was volar part of arm.
- Figure 2- Specify which image belongs to which individual and what time point (6h/12h/24h/36h/48h) post skin test. Figure legend can be changed. Instead of 'This is figure', authors can write "Representative images of DTH skin test at 6h, 12h, 24h, 36h, and 48h......". This can be Figure 2A. For Figure 2B, authors can show all 28 participants' 12h time point skin test data measured after dose 1 on day 20 and after the 2nd dose on day 36. They can present that data in the line or bar format (DTH reaction vs Days) and mark it as Figure 2B. This will support their skin test results (Line 58-60).
- Please check for spelling and punctuation errors, and variation in font and image sizes at some places.
Author Response
The present short report is a clear, concise, and well-written manuscript. The introduction is relevant. The results are compelling. Authors should fix minor punctuation errors. Overall, the authors report the use of a simple in vivo method to monitor T-cell immune responses in mRNA-SARS-Cov-2 vaccinated individuals. This simple serum test reports a remarkable increase in antibody titer post 35 days of 2 doses of vaccination and all 28 subjects developed positive DTH reaction after 2 doses. This testing method could be easily used for large-scale population vaccine studies.
Authors have already validated the DTH skin test to study the cellular immune response in COVID19 patients in their recently published paper (Barrios et al. 2021). Here, in this short communication, authors have used the same DTH skin reaction method to 2 doses of BNT162b2 mRNA vaccine in the participants to assess cellular immune response. An understanding of cellular immune response is critical in controlling infection, particularly in the case of COVID19 pandemic. Because of its affordability and simplicity, DTH skin tests could be easily used to determine the immunogenicity of newly developed vaccines in large population studies.
Authors can improve their paper by checking on the following:
Introduction can be improvised. In the beginning, authors can give an overview of mRNA-based vaccines, why mRNA vaccine platform is urgently needed and could be a promising alternative to conventional ones, further documenting its safety and efficacy.
The introduction has been redone in order to better introduce the objective of the study
Introduction
Ongoing studies are monitoring immune responses elicited by mRNA vaccines. Few studies have been focused on the role that the immune cellular responses might play in the development of immunity to SARS-CoV-2 and what are the implications for vaccines, and only humoral responses have been sistematically explored, assuming that Individuals with high antibody levels after infection have been shown to have a high number of SARS-CoV-2 specific T cells. However, there are evidences that T cells producing interferon γ have also been detected a median of 75 days after PCR confirmed covid-19 in people with undetectable SARS-CoV-2 antibodies [1], suggesting immunity is partly mediated and maintained by memory T cells. Finally, a preprint of a recent study of 100 people with a history of asymptomatic or mild covid-19 reports T cell mediated immune responses lasting for at least six months in all participants [2].
Most of these studies have been mainly focused on the humoral side with few studies addressing cellular immune responses [3-5]. One of the reasons behind this lack of studies is that both ELISAs antibody methods and in vitro cellular assays require the extraction of a blood sample from the patient that complicate massive analysis in large populations. Here we present the results of a novel application of an in vivo method to monitor cellular immune responses elicited in vaccinated individuals. This method has been previously validated in SARS-CoV-2 exposed individuals [6] showing a strong concordance (84.3%) with anti-RBD IgG.
Materials and Methods, Skin Test- Here, authors should mention the injection site. In their cited reference, it was volar part of arm.
Following the reviewer comment, this section has been already reconstructed to better describe Methods.
Participants- Twenty eight individuals were enrolled from the health care workers priority group vaccinated at the Hospital Universitario de Canarias during February-March 2021. These patients were to be vaccinated with the with BNT162b2 mRNA Pfizer vaccine. All of them were given a written document with the relevant and necessary information to decide on their participation in the study. The study is conducted in accordance with the requirements expressed in Law and the protocol was approved by the ethical committee of the Hospital (CHUC_2021_04)
Serology- The vaccine administration schedule was carried out on days 0 and 21. Serum samples were collected from the participants on days 0, 10, 20 (just before the second dose) and on day 36 (15 days after the second dose). Every single sample was sent to the Immunology laboratory and freezed. and sent to the Immunology laboratory. A commercial ELISA IgG specific for the S1 protein of SARS-CoV-2 was used according to manufacturer´s instructions (Euroimmun, Lübeck, Germany). Results were expressed as Optical Density (O.D.) ratios as as it has been recently described and validated for these determinations [7]. OD ratios under 0.8 were considered negative.
Delayed-type hypersensitivity (DTH) Skin Test - Protocol was performed according to usual clinical practice [8] and as previously described [4]. Briefly, after signing the informed consent, 25 microL (0.1 mg/mL final concentration) for intradermal puncture (IDT) of the spike protein was administered in the volar part of arm. A lyophilized recombinant SARS-CoV-2 protein of the receptor binding domain (RBD) was re-suspended in sterile water and sterile filtered 0.22 µm with a final concentration of 0.1 mg / ml following the manufacturer's instructions and under controlled sterilization conditions. The final concentration was obtained from that normally used in the tuberculin test [9]. The participants had been instructed to avoid antihistamines and corticosteroids at least 5 days before the DTH skin test and to take a well-focused photograph with a ruler to assess the size of the reaction at 6h, 12h, 24h, and 48 h after injection. DTH was performed in all patients on day 36, 14 days after the second dose. A 24-hour helpline was provided for their consultation and evaluation if necessary. A positive response was considered in the case of a positive cellular response function. Additionally, DTH was also performed in 11 patients on day 20. Ten negative controls without previous infection and without vaccination were used to see the null responsiveness of the protein.
Figure 2- Specify which image belongs to which individual and what time point (6h/12h/24h/36h/48h) post skin test. Figure legend can be changed. Instead of 'This is figure', authors can write "Representative images of DTH skin test at 6h, 12h, 24h, 36h, and 48h......". This can be Figure 2A. For Figure 2B, authors can show all 28 participants' 12h time point skin test data measured after dose 1 on day 20 and after the 2nd dose on day 36. They can present that data in the line or bar format (DTH reaction vs Days) and mark it as Figure 2B. This will support their skin test results (Line 58-60).
Following the reviewer comment, we have added a table with diameters of DTH skin reaction in all 28 individuals at 6h, 12h, 24h, and 48h after intradermal test. Further, we have added a Figure 2b, as reviewer also suggested, with five individuals with DTH on day 20 and day 36.
Please check for spelling and punctuation errors, and variation in font and image sizes at some places.
It has been corrected as reviewer has correctly pinted out.
Round 2
Reviewer 1 Report
This is round 2 of peer review of this manuscript, where the authors show that a skin hypersensitivity test detects reactivity against SARS-CoV-2 in 28 out of 28 individuals that received two doses of the Pfizer vaccine.
Abstract
“Skin Test using using S protein was” -> “using” is repeated (typo)
Introduction
“Few 29 studies have been focused on the role that the immune cellular responses might play in 30 the development of immunity to SARS-CoV-2 and what are the implications for vaccines, 31 and only humoral responses have been sistematically explored, assuming that Individu- 32 als with high antibody levels after infection have been shown to have a high number of 33 SARS-CoV-2 specific T cells.” – sentence is too long, plus it doesn’t make sense to say “assuming that individuals…have been shown”, either it’s assumed or it’s been show. Please rewrite.
“there are evidences that” -> there is evidence that (“evidence” is an uncountable noun, there’s no plural)
“vaccinated with the with BNT162b2” -> “with” is repeated (typo)
“freezed” -> frozen (“freezed” does not exist)
Figure 1 – I still don’t know what OD means in terms of IgG titer, but the fact that many publications simply report “arbitrary units” means it’s fine, it’s been done before
I can’t see Table 1 neither online (Chrome) nor on “Preview” on my computer, I hope this is not an issue in the actual manuscript final files. Also, Table legends are put on top, not on the bottom of tables (unlike figure legends, which go on the bottom), please correct that
Very good overall, all my comments are minor corrections/typos, it will be ready for publication after one more round of minor corrections in my opinion.
Author Response
All issues have been answered and corrected following reviewer recommendations.